# A Uniformly Consistent Estimator of non-Gaussian Causal Effects Under the *k*-Triangle-Faithfulness Assumption

**Shuyan Wang**                                                          SHUYANW@ANDREW.CMU.EDU
*Carnegie Mellon University*

**Peter Spirtes**                                                          PS7Z@ANDREW.COM.EDU
*Carnegie Mellon University*

**Editors:** Bernhard Schölkopf, Caroline Uhler and Kun Zhang

## Abstract

Kalisch and Bühlmann (2007) showed that for linear Gaussian models, under the Causal Markov Assumption, the Strong Causal Faithfulness Assumption, and the assumption of causal sufficiency, the PC algorithm is a uniformly consistent estimator of the Markov Equivalence Class of the true causal DAG for linear Gaussian models; it follows from this that for the identifiable causal effects in the Markov Equivalence Class, there are uniformly consistent estimators of causal effects as well. The ***k*-Triangle-Faithfulness Assumption** is a strictly weaker assumption that avoids some implausible implications of the Strong Causal Faithfulness Assumption and also allows for uniformly consistent estimates of Markov Equivalence Classes (in a weakened sense), and of identifiable causal effects (Spirtes and Zhang, 2016). However, both of these assumptions are restricted to linear Gaussian models. We propose the Generalized $k$-Triangle Faithfulness, which can be applied to any smooth distribution. In addition, under the Generalized $k$-Triangle Faithfulness Assumption, we describe the Edge Estimation Algorithm that provides uniformly consistent estimators of causal effects in some cases (and otherwise outputs "can't tell"), and the *Very Conservative SGS* Algorithm that (in a slightly weaker sense) is a uniformly consistent estimator of the Markov equivalence class of the true DAG.

**Keywords:** Machine Learning, Causal Inference, Graphical Modeling

## 1. Introduction

It has been proved that under the Causal Markov, Faithfulness assumptions and Causal Sufficiency Assumption, there are no uniformly consistent estimators of Markov equivalence classes of causal structures represented by directed acyclic graphs (DAG)(Robins et al., 2003). Kalisch and Bühlmann (2007) showed that for linear Gaussian models, under the Causal Markov Assumption, the Strong Causal Faithfulness Assumption, and the assumption of causal sufficiency, the PC algorithm is a uniformly consistent estimator of the Markov Equivalence Class of the true causal DAG for linear Gaussian models; it follows from this that for the identifiable causal effects in the Markov Equivalence Class, there are uniformly consistent estimators of causal effects as well. The ***k*-Triangle-Faithfulness Assumption** is a strictly weaker assumption that avoids some implausible implications of the Strong Causal Faithfulness Assumption and also allows for uniformly consistent estimates of Markov Equivalence Classes (in a weakened sense), and of identifiable causal effects.

However, both of these assumptions are restricted to linear Gaussian models. We propose the Generalized $k$-Triangle Faithfulness, which can be applied to any smooth distribution. In addition, under the Generalized $k$-Triangle Faithfulness Assumption, we describe the Edge Estimation Algorithm that provides uniformly consistent estimates of causal effects in some cases (and otherwise

outputs "can't tell"), and the Very Conservative SGS Algorithm that (in a slightly weaker sense) is a uniformly consistent estimator of the Markov equivalence class of the true DAG. The contribution of our work is that it shows that given uniform consistency of nonparametric conditional independence tests (which itself requires a smoothness assumption), we can reach uniform consistency of estimating causal structures and effects without making parametric assumptions. The only uniform consistency results for causal search and estimation algorithms that we know of (Kalisch and Bühlmann, 2007), (Spirtes and Zhang, 2014) assume linear Gaussian models. Since uniform consistency allows one to give probabilistic bounds on the size of errors given a sample size as a function of $k$ and the smoothness parameter, this gives some small sample results for algorithms that only currently have large sample guarantees.

## 2. The Basic Assumptions for Causal Discovery

### 2.1. DAG and Causal Markov Condition

We use directed acyclic graphs to represent causal relations between variables. A directed graph $G = \langle \mathbf{V}, \mathbf{E} \rangle$ consists of a set of *nodes* $\mathbf{V}$ and a set of *edges* $\mathbf{E} \subset \mathbf{V} \times \mathbf{V}$. If there is an edge $\langle A, B \rangle \in \mathbf{E}$, we write $A \to B$. $A$ is a **parent** of $B$, and $B$ is a **child** of $A$, the edge is out of $A$ and into $B$, and $A$ is the source and $B$ is the target. A **directed path** from $X$ to $Y$ is a sequence of ordered edges where the source of the first edge is $X$, the target of the last edge is $Y$, and if there are n edges in the sequence, for $1 \leq i < n$, the target of the $i$th edge is the source of the $i + 1$th edge; $X$ is an **ancestor** of $Y$, and $Y$ is a **descendant** of $X$.

If a variable $Y$ is in a structure $X \to Y \leftarrow Z$, and there is no edge between $X$ and $Z$, we call $\langle X, Y, Z \rangle$ an *unshielded collider*; if there is also an edge between $X$ and $Z$, then $\langle X, Y, Z \rangle$ is a triangle and we call $\langle X, Y, Z \rangle$ a *shielded collider*. If $\langle X, Y, Z \rangle$ is a triangle but $Y$ is not a child of both $X$ and $Z$, we call $\langle X, Y, Z \rangle$ a *shielded non-collider*; if there is no edge between $X$ and $Z$, then $\langle X, Y, Z \rangle$ is a *unshielded non-collider.*

A *Bayesian network* is an ordered pair $\langle P, G \rangle$ where $P$ is a probability distribution over a set of variables $\mathbf{V}$ in $G$. A distribution $P$ over a set of variables $\mathbf{V}$ satisfies the **(local) Markov condition** for $G$ if and only if each variable in $\mathbf{V}$ is independent of its non-parents and non-descendants, conditional on its parents. Given $M = \langle P, G \rangle$, $P_M$ denotes $P$ and $G_M$ denotes $G$. Two acyclic directed graphs (DAG) $G_1$ and $G_2$ are *Markov equivalent* if conditional independence relations entailed by Markov condition in $G_1$ are the same as in $G_2$. It has been proven that two $DAG$s are *Markov equivalent* if and only if they have the same adjacencies and same *unshielded colliders* (Verma and Pearl, 1990). A *pattern* $O$ is an undirected graph that represents a set $M$ of Markov equivalent DAGs: an edge $X \to Y$ is in $O$ if it is in every DAG in $M$; if $X \to Y$ is in some DAG and $Y \to X$ in some other DAG in $M$, then $X - Y$ in $O$ (Spirtes and Zhang, 2016)

We assume **causal sufficiency**, which means that $\mathbf{V}$ contains all direct common causes of variables in $\mathbf{V}$.

### 2.2. Faithfulness, linear Gaussian case and *k*-Triangle-Faithfulness

Given a $\langle P, G \rangle$ that satisfies **Markov Condition**, we say that $P$ is *faithful* to $G$ if any conditional independence relation that holds in $P$ is entailed by $G$ by the **Markov Condition**. We further make the Causal Markov and Faithfulness assumption (Spirtes et al., 2001):

**Causal Markov Assumption:** If the true causal model $M$ of a population is causally sufficient, each variable in $V$ is independent of its non-parents and non-descendants, conditional on its parents in $G_M$ (Spirtes and Zhang, 2016).

**Causal Faithfulness Assumption:** all conditional independence relations that hold in the population are consequences of the Markov condition from the underlying true causal DAG.

In this paper we talk about cases where $P_M$ over $\mathbf{V}$ for $G = \langle \mathbf{V}, \mathbf{E} \rangle$ respects the **Causal Markov Assumption**. If $P_M$ is *faithful* to $G_M$ and all variables in $M$ are Gaussian and all causal relations are linear, that is, any $X_i \in \mathbf{V}$ can be written as:

$$X_i = \sum_{X_j \in Pa_M(X_i)} a_{i,j} X_j + \epsilon_j$$

where $Pa_M(X)$ denotes the set of parents of $X$ in $G_M$, $a_{i,j}$ is a real valued coefficient, and the set of $\epsilon_j$ are multivariate Gaussian and jointly independent, conditional correlation between any two variables $\rho_{X,Y|\mathbf{A}} = 0$ where $X, Y \in \mathbf{V}$ and $\mathbf{A} \subset \mathbf{V} \setminus \{X, Y\}$ implies that there is no edge between $X$ and $Y$. Based on the equation above, we define in the linear Gaussian case the *edge strength* $e_M(X_j \to X_i)$ as the corresponding coefficient $a_{i,j}$.

It has been proved that under the Causal Markov and Faithfulness assumptions, there are no uniformly consistent estimators of Markov equivalence classes of causal structures represented by DAG (Robins et al., 2003). Kalisch and Bühlmann (2007) showed that such uniform consistency is achieved by the $PC$ algorithm if the underlying DAG is sparse relative to the sample size under a strengthened version of Faithfulness Assumption; in particular, with $p$ denoting the number of nodes in the DAG, $q$ the maximal degree of the DAG and $n$ the sample size, they assumed that $p = \mathcal{O}(n^a)$ for some $0 \le a \le \infty$ and $q = \mathcal{O}(n^{1-b})$ for some $0 < b < 1$. This **Strong Causal Faithfulness Assumption** in the linear Gaussian case bounds the absolute value of any partial correlation not entailed to be zero by the true causal DAG away from zero by some positive constants. It has the implausible consequence that it puts a lower bound on the strength of edges, since a very weak edge entails a very weak partial correlation. However, the **Strong Causal Faithfulness Assumption** can be weakened to the strictly weaker (for some values of $k$) **$k$-Faithfulness Assumption** while still achieving uniform consistency. Furthermore, at the cost of having a smaller subset of edges oriented, the **$k$-Faithfulness Assumption** can be weakened to the **$k$-Triangle-Faithfulness Assumption**, while still achieving uniform consistency and can be relaxed while preserving the uniform consistency: the **$k$-Triangle-Faithfulness Assumption** (Spirtes and Zhang, 2014) only bounds the conditional correlation between variables in a triangle structure from below by some functions of the corresponding edge strength:

**$k$-Triangle-Faithfulness Assumption:** Given a set of variables $\mathbf{V}$, suppose the true causal model over $\mathbf{V}$ is $M = \langle P, G \rangle$, where $P$ is a Gaussian distribution over $\mathbf{V}$, and $G$ is a DAG with vertices $\mathbf{V}$. For any variables $X, Y, Z$ that form a triangle in $G$:

- if $Y$ is a non-collider on the path $\langle X, Y, Z \rangle$, then $|\rho_M(X, Z|\mathbf{W})| \ge k \times |e_M(X - Z)|$ for all $\mathbf{W} \subset \mathbf{V}$ that do not contain Y; and

- if $Y$ is a collider on the path $\langle X, Y, Z \rangle$, then $|\rho_M(X, Z|\mathbf{W})| \ge k \times |e_M(X - Z)|$ for all $\mathbf{W} \subset \mathbf{V}$ that do contain Y.

where the $X - Z$ represents the edge between $X$ and $Z$ but the direction is not determined.(Sprites and Zhang, 2014)

The **k-Triangle-Faithfulness Assumption** is strictly weaker than the **Strong Faithfulness Assumption** in several respects: the **Strong faithfulness Assumption** does not allow edges to be weak any where in a graph, while the **k-Triangle-Faithfulness Assumption** only excludes conditional correlations $\rho(X, Z|\mathbf{W})$ from being too small if $X$ and $Z$ are in some triangle structures $\langle X, Y, Z \rangle$ and $X - Z$ is not a weak edge; for every $\epsilon$ used in the **Strong Faithfulness Assumption** as the lower bound for any partial correlation, there is a $k$ for the **k-Triangle-Faithfulness Assumption** that gives a lower bound smaller than $\epsilon$.

### 2.3. VCSGS Algorithm

The algorithm we use to infer the structure of the underlying true causal graph is *Very Conservative SGS* ($VCSGS$) algorithm, which uses uniformly consistent tests of conditional independence.

**VCSGS Algorithm**

1. Form the complete undirected graph $H$ on the given set of variables $\mathbf{V}$.

2. For each pair of variables $X$ and $Y$ in $\mathbf{V}$, search for a subset $\mathbf{S}$ of $\mathbf{V} \setminus \{X, Y\}$ such that $X$ and $Y$ are independent conditional on $\mathbf{S}$. Remove the edge between $X$ and $Y$ in $H$ if and only if such a set is found.

3. Let $K$ be the graph resulting from Step 2. For each unshielded triple $\langle X, Y, Z \rangle$ (the only pair of variables not adjacent are $X$ and $Z$ ),

   (a) If $X$ and $Z$ are not independent conditional on any subset of $\mathbf{V} \setminus X, Z$ that contains $Y$, then orient the triple as a collider: $X \rightarrow Y \leftarrow Z$.

   (b) If $X$ and $Z$ are not independent conditional on any subset of $\mathbf{V} \setminus X, Z$ that does not contain Y, then mark the triple as a non-collider.

   (c) Otherwise, mark the triple as ambiguous.

4. Execute the following orientation rules until none of them applies:

   (a) If $X \rightarrow Y - Z$, and the triple $\langle X, Y, Z \rangle$ is marked as a non-collider, then orient $Y - Z$ as $Y \rightarrow Z$.

   (b) If $X \rightarrow Y \rightarrow Z$ and $X - Z$, then orient $X - Z$ as $X \rightarrow Z$.

   (c) If $X \rightarrow Y \leftarrow Z$, another triple $\langle X, W, Z \rangle$ is marked as a non-collider, and $W - Y$, then orient $W - Y$ as $W \rightarrow Y$.

5. Let $M$ be the graph resulting from step 4. For each consistent disambiguation of the ambiguous triples in $M$ (*i.e.*, each disambiguation that leads to a pattern), test whether the resulting pattern satisfies the Markov condition. If every pattern does, then mark all the 'apparently non-adjacent' pairs as 'definitely non-adjacent'.

It has been proved that under the **k-Triangle-Faithfulness Assumption**, $VCSGS$ algorithm is uniformly consistent in the inference of graph structure. Furthermore, a follow-up algorithm that estimates edge strength given the output of $VCSGS$ also reaches uniform consistency. We are going to prove that the uniform consistency of the estimation of the causal influences under the **k-Triangle-Faithfulness Assumption** can be extended to discrete and nonparametric cases as long as

there are uniformly consistent tests of conditional independence (which in the general case requires a smoothness assumption), by showing that missed edges in the inference of causal structure are so weak that the estimations of the causal influences are still uniformly consistent.

Notice that $VCSGS$ is not practically feasible, which can be seen by its computation complexity of $\mathcal{O}(2^{n-2}n^2)$ where $n$ is the number of vertices in the causal graph. We are presenting this algorithm here mainly to use it to show that uniform consistency can be reached when estimating causal effects without the need of bounding all nonzero conditional correlations away from zero by a constant.

## 3. Nonparametric Case

For nonparametric case, we consider variables supported on $[0,1]$. We first define the strength of the edge $X \to Y$ as the maximum change in $L1$ norm of the probability of $Y$ when we condition on only parents of $Y$ and only change the value of $X$:

$$\text{If } x \in Pa(Y):$$
$$e(X \to Y) := \max_{pa_{\backslash \{x\}}(Y) \in [Pa(Y) \backslash \{X\}]} \max_{x_1, x_2 \in [X]} ||p_{Y|x_1, pa_{\backslash \{x\}}(Y)} - p_{Y|x_2, pa_{\backslash \{x\}}(Y)}||_1$$

where $[X]$ denotes the set of values that $X$ takes, $[Pa(Y)] \subset [0,1]^{|Pa(Y)|}$ the set of values that parents of Y take, $p_{Y|x_1, pa_{\backslash \{x\}}(Y)}$ the probability distribution of $Y|X = x_1, Pa(Y) \backslash \{X\} = pa_{\backslash \{x\}}(Y)$ and $p_{y|x_1, pa_{\backslash \{x\}}(Y)}$ the density of $p_{Y|x_1, pa_{\backslash \{x\}}(Y)}$ for $Y = y$. Since we are conditioning on the set of parents, the conditional probability is equal to the manipulated probability. Although we choose the maximum change of the distribution of $Y$ for defining the edge strength, it is not the only way to define it. For instance, the average can also be chosen to define the edge strength in which case the rest of the approach only need to be changed slightly. We choose the max because we are particularly interested in weak edges, and if an edge is weak, the maximum of change on effect when changing the state of cause is small.

Then we can make the ***k*-Triangle-Faithfulness Assumption**: given a set of variables **V**, where the true causal model over **V** is $M = \langle P, G \rangle$, $P$ is a distribution over **V**, and $G$ is a DAG with vertices **V**, for any variables $X, Y, Z$ that form a triangle in $G$:

- if $Z$ is a non-collider on the path $\langle X, Z, Y \rangle$, given any subset $\mathbf{W} \subset \mathbf{V} \backslash \{X, Y, Z\}$, $\min_{\mathbf{w} \in [\mathbf{W}]} \min_{x_1, x_2 \in [X]} ||p_{Y|\mathbf{w}, x_1} - p_{Y|\mathbf{w}, x_2}||_1 \geq K_Y e(X \to Y)$ for some $K_Y > 0$

- if $Z$ is a collider on the path $\langle X, Z, Y \rangle$, then for every $y \in [Y]$, given any subset $\mathbf{W} \subset \mathbf{V} \backslash \{X, Y, Z\}$, $\min_{\mathbf{w} \in [\mathbf{W}]} \min_{z \in [Z]} \min_{x_1, x_2 \in [X]} ||p_{Y|\mathbf{w}, x_1, z} - p_{Y=y|\mathbf{w}, x_2, z}||_1 \geq K_Y e(X \to Y)$ for some $K_Y > 0$

In order to have uniformly consistent tests of conditional independence, we make smoothness assumption for continuous variables with the support on $[0,1]$:

*TV (Total Variation) Smoothness(L):* Let $\mathcal{P}_{[0,1], TV(L)}$ be the collection of distributions $p_{Y, \mathbf{A}}$, such that for all $\mathbf{a}, \mathbf{a}' \in [0,1]^{|\mathbf{A}|}$, we have: $||p_{Y|\mathbf{A}=\mathbf{a}} - p_{Y|\mathbf{A}=\mathbf{a}'}||_1 \leq L||\mathbf{a} - \mathbf{a}'||_1$

Given the TV smoothness(L), $p$ is continuous: if $a$ and $a'$ are arbitrarily close, $|p_{Y|\mathbf{A}=\mathbf{a}} - p_{Y|\mathbf{A}=\mathbf{a}'}|$ is arbitrarily small. Furthermore, since $[0,1]^d$ ($d \in \mathbb{N}$) is compact, for any $\mathbf{W}, \mathbf{U} \subset \mathbf{V}$ (the set of all variables in the true causal graph) , $p_{U|W}$ attains its max and min on its support. Since $|\mathbf{V}|$ is finite, we can further assume conditional densities are non-zero (NZ(T)): for any $X \in \mathbf{V}, \mathbf{U} \subset \mathbf{V}, p_{X|\mathbf{U}} \geq T$ for some $1 > T > 0$.

Notice that by TV Smoothness(L) and that variables have support on $[0, 1]$, we can derive an upper bound for probability of any variable given its parents, so $p_{Y|Pa(Y)}$ cannot be infinitely large:
$$p_{Y|Pa(Y)=pa(Y)} \leq ||p_{Y|Pa(Y)=pa(Y)}||_1 \leq ||p_{Y|Pa(Y)=pa'(Y)}||_1 + L||pa(Y) - pa'(Y)||_1 \leq (1 + L|Pa(Y)|)$$ $(|Pa(Y)|)$ refers to the number of parents of $Y$)

Although the discrete probability case does not have support on $[0, 1]$, and its probability is not continuous, it still satisfies the TV smoothness(L) assumption: for instance, if the discrete variables have support only on integers, we can set $L = 1$. By replacing the density $p_{X|\mathbf{U}}$ with the probability $P(X|\mathbf{U})$ in NZ(T) assumption, we have a NZ(T) assumption for the discrete case. Therefore the proof of uniform consistency for the nonparametric case in the rest of the paper also works for the discrete case.

### 3.1. Uniform Consistency in the inference of structure

We use $L1$ norm to characterize dependence over all states of $X$ ,$Y$ and $Z$: $\epsilon_{X,Y|\mathcal{A}} = ||p_{X,Y,\mathcal{A}} - p_{X|\mathcal{A}}p_{Y|\mathcal{A}}p_{\mathcal{A}}||_1$. We want a test $\psi$ of $H_0 : \epsilon = 0$ versus $H_1 : \epsilon > 0$. $\psi$ is a family of functions: $\psi_0...\psi_n...$ one for each sample size, that takes an i.i.d sample $V_n$ from the joint distribution over $\mathbf{V}$. Then the test is uniformly consistent w.r.t. a set of distributions $\Omega$ if :

- $\lim_{n\to\infty} \sup_{P \in \mathcal{P}_{[0,1],TV(L)},\epsilon(P)=0} P^n(\psi_n(V_n) = 1) = 0$

- for every $\delta > 0$, $\lim_{n\to\infty} \sup_{\epsilon(P)\geq\delta} P^n(\psi_n(V_n) = 0) = 0$

With the TV Smoothness(L) assumption, there are uniformly consistent tests of conditional independence, such as a minimax optimal conditional independence test proposed by Neykov et al.(2020).

Given any causal model $M = \langle P, G \rangle$ over $\mathbf{V}$, let $C(n, M)$ denote the (random) output of the $VCSGS$ algorithm given an $i.i.d.$ sample of size $n$ from the distribution $P_M$, then there are three types of errors that it can contain that will mislead the estimation of causal influences:

1. $C(n, M)$ *errs in kind I* if it has an adjacency not in $G_M$;

2. $C(n, M)$ *errs in kind II* if every adjacency it has is in $G_M$, but it has a marked non-collider not in $G_M$;

3. $C(n, M)$ *errs in kind III* if every adjacency and marked non-collider it has is in $G_M$, but it has an orientation not in $G_M$

If $C(n, M)$ errs in either of these three way, there will be variable $X$ and $Y$ in $C(n, M)$ such that $X$ is treated as a parent of $Y$ but is not in the true graph $G_M$; if there is no undirected edge connecting $Y$ in this $C(n, M)$, the algorithm will estimate the causal influence of " parents" of $Y$ on $Y$, but such estimation does not bear useful information since intervening on $X$ does not really influence $Y$. Notice that missing an edge is not listed as an mistake here, and we are going to prove later that the estimation of causal influence can still be used to correctly predict the effect of intervention even if the algorithm misses edges. This is because an edge is only going to be missed if it is a weak edge; and if such an edge is missed, we still can achieve uniform consistency when estimating $p_{Y|Pa'(Y)}$ where here $Pa'(Y)$ is the parent of $Y$ except those connected to $Y$ through the missed, weak edges.

Now we present the uniform consistency result of the inference of causal structure:

**Theorem 1** *Let $\phi^{k,L,T}$ be the set of causal models over **V** under **k-Triangle-Faithfulness Assumption**, TV smoothness(L) and the assumptions of NZ(T). Under the causal sufficiency of the measured variables **V**, causal Markov assumption, $k$-Triangle-Faithfulness, TV smoothness(L) assumption and NZ(T) assumption,*

$$\lim_{n \to \infty} \sup_{M \in \phi^{k,L,C}} P_M^n(C(n,M)errs) = 0$$

Since the error of the algorithm exists in the deciding the whether certain edge exists and it direction, and the algorithm makes such decision based on the result conditional Independence test, in order to find the limit of upper bound of the algorithm error, we need to first find the relation between the conditional dependence between two variables and the strength of the edge that connects them. Lemma bounds $\epsilon_{X,Y|\mathbf{A}}$ with strengths of the edge $X \to Y$. All proofs not shown in the paper will be found in the appendix.

**Lemma 2**

*Given an ancestral set $\mathbf{A} \subset \mathbf{V}$ that contains the parents of $Y$ but not $Y$:*
*If $X$ is a parent of $Y$:*

$$T^{|\mathbf{A}|}e(X \to Y) \le \epsilon_{X,Y|\mathbf{A}\setminus\{X\}} \le e(X \to Y)$$

We are going to prove for each case that the probability for $C(n,M)$ to make each of the three kinds of mistakes uniformly converges to zero. Since the proofs for the *kind I* and *kind III* errors are almost the same as the proof for the linear Gaussian case (Spirtes and Zhang, 2014), we are only gong to prove *kind II*. We provide the proofs of *kind I* and *kind III* errors (Spirtes and Zhang, 2014) in the appendix.

**Lemma 3** *Given causal sufficiency of the measured variables **V**, the Causal Markov, $k$-Triangle-Faithfulness, TV smoothness(L) assumption and NZ(T) assumption:*

$$\lim_{n \to \infty} \sup_{M \in \phi^{k,L,C}} P_M^n(C(n,M) \text{ errs in kind II}) = 0$$

**Proof** For any $M \in \phi^{k,L,C}$, if it errs in kind II then it contains a marked non-collider $\langle X, Z, Y \rangle$ that is not in $G_M$. Since it's been proved (Spirtes and Zhang, 2014):
$$\lim_{n \to \infty} \sup_{M \in \phi^{k,L,C}} P_M^n(C(n,M) \text{ errs in kind I}) = 0$$
the errors of kind II can be one of the two cases:
$(I)$ $\langle X, Z, Y \rangle$ is an unshielded collider in $G_M$;
$(II)$ $\langle X, Z, Y \rangle$ is a shielded collider in $G_M$;
the proof for case (I) is the same as the proof for the $C(n,M)$ errs in kind I (Spirtes and Zhang, 2014), so we are going to prove here that the probability of case (II) uniformly converges to zero as sample size increases.

We are going to prove by contradiction. Suppose that the probability that $VCSGS$ making a mistake of kind II does not uniformly converge to zero. Then there exists $\lambda > 0$, such that for every sample size $n$, there is a model $M(n)$ such that the probability of $C(n, M(n))$ containing an unshielded non-collider that is a shielded collider in $M(n)$ is greater than $\lambda$. Let that triangle be $\langle X^{M(n)}, Z^{M(n)}, Y^{M(n)} \rangle$ with $X^{M(n)}$ being the parent of $Y^{M(n)}$ in $M(n)$.

The algorithm will identify the triple as an unshielded non-collider only if:

$(i)$ there is a set $\mathbf{U}^{M(n)} \subset \mathbf{V}^{M(n)}$ containing $Z^{M(n)}$, such that the test of $\epsilon_{X^{M(n)}, Y^{M(n)} | \mathbf{U}^{M(n)}} = 0$ returns 0, call this test $\psi_{n0}$;

$(ii)$ there is an ancestral set $\mathbf{W}^{M(n)}$ that contains $X^{M(n)}$ and $Y^{M(n)}$ but not $Z^{M(n)}$, such that for set $\mathbf{A}^{M(n)} = \mathbf{W}^{M(n)} \setminus \{X^{M(n)}, Y^{M(n)}\}$, the test $\epsilon_{X^{M(n)}, Y^{M(n)} | \mathbf{A}^{M(n)}} = 0$ returns 1, call this test $\psi_{n1}$.

If what we want to proof does not hold for the algorithm, for all $n$ there is a model $M(n)$:

$(1) P^n_{M(n)}(\psi_{n0} = 0) > \lambda$

$(2) P^n_{M(n)}(\psi_{n1} = 1) > \lambda$

(1) tells us that there is some $\delta_n$ such that $|\epsilon_{X^{M(n)}, Y^{M(n)} | \mathbf{U}^{M(n)}}| < \delta_n$ and $\delta_n \to 0$ as $n \to \infty$ since the test is uniformly consistent. So we have:

$$\delta_n > \epsilon_{X^{M(n)}, Y^{M(n)} | \mathbf{U}^{M(n)}} \tag{1}$$

$$= \mathbb{E}_{\mathbf{U}^{M(n)} \sim p_{\mathbf{U}^{M(n)}}} \left[ \int_{X^{M(n)}} p_{x^{M(n)} | \mathbf{U}^{M(n)}} ||p_{Y^{M(n)} | x^{M(n)}, \mathbf{U}^{M(n)}} - p_{Y^{M(n)} | \mathbf{U}^{M(n)}}||_1 dx^{M(n)} \right] \tag{2}$$

$$\geq \min_{x_1^{M(n)}, x_2^{M(n)} \in [X^{M(n)}]} ||p_{Y^{M(n)} | x_1^{M(n)}, \mathbf{U}^{M(n)}} - p_{Y^{M(n)} | x_2^{M(n)}, \mathbf{U}^{M(n)}}||_1 \tag{3}$$

$$\geq K_{Y^{M(n)}} e_M(X^{M(n)} \to Y^{M(n)}) \tag{4}$$

The last step is by $k$-Triangle-Faithfulness

So $e_M(X^{M(n)} \to Y^{M(n)}) < \dfrac{\delta_n}{K_{Y^{M(n)}}}$.

By Lemma 2, $\epsilon_{X^{M(n)}, Y^{M(n)} | \mathbf{A}^{M(n)}} < e_M(X^{M(n)} \to Y^{M(n)})$.

Therefore, $\epsilon_{X^{M(n)}, Y^{M(n)} | \mathbf{A}^{M(n)}} < \dfrac{\delta_n}{K_{Y^{M(n)}}} \to 0$ as $n \to \infty$, which violates the condition $(ii)$, which says that the uniformly consistency test will reject that $\epsilon_{X^{M(n)}, Y^{M(n)} | \mathbf{A}^{M(n)}} = 0$. Contradiction. ∎

**Theorem 4** *Given causal sufficiency of the measured variables* $\mathbf{V}$, *the Causal Markov, $k$-Triangle-Faithfulness, TV smoothness(L) and NZ(T) assumptions:*

$$\lim_{n \to \infty} \sup_{M \in \phi^{k,L,C}} P^n_M(C(n, M) \text{ errs}) = 0$$

**Proof** Since we have proved that the probability for $C(n, M)$ to make any of the three kinds of mistakes uniformly converges to 0, the theorem directly follows. ∎

### 3.2. Uniform consistency in the inference of causal effects

**Edge Estimation Algorithm (Sprites and Zhang, 2014):**

1. Run the VCSGS algorithm on an i.i.d sample of size $n$ from $P_M$.

2. Let the output from step 1 be $C(n, M)$.

3. If some non-adjacencies in $C(n, M)$ are not marked as 'definitely non-adjacent', return 'Unknown' for every pair of variables.

4. If all non-adjacencies in $C(n, M)$ are marked as 'definitely non-adjacent', then:

   (a) For a vertex $Y$ such that all of the edges containing $Y$ are oriented in $C(n, M)$, if $Pa(Y)$ is the parent set of $Y$ in $C(n, M)$, use histogram to estimate $p(y_i|Pa(Y) = pa(Y))$[1]for $y_i \in [Y]$ and $pa(Y) \in [Pa(Y)]$;

   (b) for any of the remaining edges, return 'Unknown';

**Defining the distance between $M_1$ and $M_2$**

The method for estimation for $p(y|Pa(Y) = pa(Y))$ is: we first get $\hat{p}(Y = y, Pa(Y) = pa(Y))$ and $\hat{p}(Pa(Y) = pa(Y))$ by histogram, then we get:

$$\hat{p}(y|Pa(Y) = pa(Y)) = \frac{\hat{p}(Y = y, Pa(Y) = pa(Y))}{\hat{p}(Pa(Y) = pa(Y))}$$

Let $M_1$ be the output of the Edge Estimation Algorithm, and $M_2$ be a causal model, we define the *conditional probability distance*, $d[M_1, M_2]$, between $M_1$ and $M_2$ to be:

$$d[M_1, M_2] = \max_{\substack{Y \in \mathbf{V}, \\ y_i \in [Y], \\ pa_{M_1}(Y) \\ \in [Pa_{M_1}(Y)], \\ pa_{M_2}(Y) \\ \in [Pa_{M_2}(Y)], \\ pa_{M_1} \subset pa_{M_2}}} |\hat{p}_{M_1}(y_i|pa_{M_1}(Y)) - p_{M_2}(y_i|pa_{M_2}(Y))|$$

where $Pa_M(Y)$ denotes the parent set of $Y$ in causal model $M$. The intuition behind this definition of *conditional probability distance* is that among all variables $Y$ in $M_1$ that satisfy step 4(a) in the Edge Estimation Algorithm, each of them will have an estimated conditional probability given each different state of their parents in $M_1$; the *conditional probability distance* captures the maximum difference between the estimated conditional probability of a variable $\hat{p}_{M_1}(y_i|pa_{M_1}(Y))$ and its true conditional probability in $M_2$, $p_{M_2}(y_i|pa_{M_2}(Y))$, where all the parents that are also in $Pa_{M_1}(Y)$ are in the same state as $pa_{M_1}(Y)$. By convention $|\hat{P}_{M_1}(y_i|pa_{M_1}(Y)) - P_{M_2}(y_i|pa_{M_2}(Y))| = 0$ if $\hat{P}_{M_1}(y_i|pa_{M_1}(Y))$ is "Unknown".

Now we want to show, the edge estimation algorithm is uniformly consistent.

**Theorem 5** *for every $\delta > 0$,*

$$\lim_{n \to \infty} \sup_{M \in \phi^{k, L, C}} P_M^n(d[\hat{O}(M), M] > \delta) = 0$$

*Here $M$ is any causal model satisfying causal sufficiency of the measured variables $\mathbf{V}$, the Causal Markov, k-Triangle-Faithfulness, TV smoothness(L) and NZ(T) assumptions and $\hat{O}(M)$ is the output of the algorithm given an iid sample from $P_M$.*

**Proof** Let $\mathcal{O}$ be the set of possible graphs of $VCSGS$. Since given $\mathbf{V}$, there are only finitely many outputs in $\mathcal{O}$, it suffices to prove that for each output $O \in \mathcal{O}$,

---

1. we denote the density of $p_{Y|Pa(Y)=pa(Y)}$ at $Y = y$ as $p(y|Pa(Y) = pa(Y))$ in this section to match with the result of estimation.

$$\lim_{n\to\infty} \sup_{M\in\phi^{k,L,C}} P_M^n(d[\hat{O}(M), M] > \delta | C(n, M) = O)P_M^n(C(n, M) = O) = 0$$

Now partition all the $M$ into three sets given $O$ :

- $\Psi_1 = \{M|$ all adjacencies, non-adjacencies and orientations in O are true in $M\}$;

- $\Psi_2 = \{M|$ only some adjacencies, or orientations in O are not true in $M\}$;

- $\Psi_3 = \{M|$ only some non-adjacencies in O are not true in $M\}$.

It suffices to show that for each $\Psi_i$,

$$\lim_{n\to\infty} \sup_{M\in\Psi_i} P_M^n(d[\hat{O}(M), M] > \delta | C(n, M) = O)P_M^n(C(n, M) = O) = 0$$

$\Psi_1$:

For any $M \in \Psi_1$, if the conditional probabilities of a vertex $Y$ in $O$ can be estimated (so not "Unknown"), it means that $Pa_O(Y) = Pa_M(Y)$. Recall that the histogram estimator is close to the true density with high probability:

for any $\lambda < 1$, $\sup_{P\in\mathcal{P}_{TV(L)}} P^n(||\hat{p}_h(x) - p(x)||_\infty \le f(n, \lambda) \le \mathcal{O}((\frac{\log n}{n})^{\frac{1}{2+d}})) \ge 1 - \lambda$

where $f(n, \lambda)$ is continuous and monotonically decreasing wrt $n$ and $\epsilon$ and $h \propto n^{2/(2+d)}$ where $d$ is the dimentionality of the $x$ and $h$ is the size of bins or sub-cubes values of variables are divided into to estimate probability. For instance, $d = |Pa(Y)| + 1$ when estimating $P(Y, Pa(Y))$.

Given a $\delta > 0$, $f(n, \epsilon) = \delta$ entails that $\sup_{P\in\mathcal{P}_{[0,1],TV(L)}} P^n(||\hat{p}_h(x) - p(x)||_\infty > \delta) < \lambda$.

By monotonicity of $f(n, \lambda)$, when $n > n_f$ s.t. $f(n_f, \lambda) = \delta$, $\sup_{P\in\mathcal{P}_{[0,1],TV(L)}} P^n(||\hat{p}_h(x) - p(x)||_\infty > \delta) < \lambda$.

Therefore the histogram estimators of $p_M(y, Pa_M(Y) = pa_M(Y))$ and $p_M(Pa_M(Y) = pa_M(Y))$ are uniformly consistent.

**Lemma 6** *If $\hat{p}(Y = y, Pa(Y) = pa(Y))$ and $\hat{p}(Pa(Y) = pa(Y))$ are uniformly consistent estimators of $p(Y = y, Pa(Y) = pa(Y))$and $p(Pa(Y) = pa(Y))$, then*

$$\hat{p}(y|Pa(Y) = pa(Y)) = \frac{\hat{p}(Y = y, Pa(Y) = pa(Y))}{\hat{p}(Pa(Y) = pa(Y))}$$

*is a uniformly consistent estimator for $p(y|Pa(Y) = pa(Y))$*

By lemma 6, we conclude that the $\hat{p}_M(y|Pa_M(Y) = pa_M(Y))$ is a uniformly consistent estimator for

$$p_M(y|Pa_M(Y) = pa_M(Y)) = \frac{p_M(y, Pa_M(Y) = pa_M(Y))}{p_M(Pa_M(Y) = pa_M(Y))},$$

So:

$$\lim_{n\to\infty} \sup_{M\in\Psi_1} P_M^n(d[\hat{O}(M), M] > \delta | C(n, M) = O)P_M^n(C(n, M) = O)$$

$$\le \lim_{n\to\infty} \sup_{M\in\Psi_1} P_M^n(d[\hat{O}(M), M] > \delta | C(n, M) = O) = 0$$

$\Psi_2$ : the proof is exactly the same as for the discrete case.

$\Psi_3$ :

Let $O(M)$ be the population version of $\hat{O}(M)$. Since the histogram estimator is uniformly consistent over $||\hat{p}_h - p||_\infty$ and there are finitely many parent-child combinations, for every $\lambda > 0$ there is a sample size $N_1$, such that for $n > N_1$, and all $M \in \Psi_3$,

$$P_M^n(d[\hat{O}(M), O(M)] > \delta/2 | C(n, M) = O) < \lambda$$

Since only some non-adjacencies in $O$ are not true in $M$, we know that for any vertex $Y$ that have some estimated conditional probabilities given its parents in $O$, $Pa_{O(M)}(Y) \subset Pa_M(Y)$ where $Pa_{O(M)}(Y)$ denotes the parent set of $Y$ in the $O$ when the underlying probability is $P_M$(i.e., $M$ is the true causal model). Since $Pa_M(Y) \not\subset Pa_{O(M)}(Y)$, for any $y_i \in [Y]$ and $pa_{O(M)}(Y) \in [Pa_{O(M)}(Y)]$, $P_O(y_i|Pa_{O(M)}(Y) = pa_{O(M)}(Y))$ is a marginalization of $p_M(y_i|Pa_M(Y) = pa_M(Y))$. Therefore, the distance between $O(M)$ and $M$ is:

$$d[O(M), M] = \max_{\substack{Y \in \mathbf{V}, \\ y_i \in [Y], \\ pa_{O(M)}(Y) \in \\ [Pa_{O(M)}(Y)], \\ pa_M(Y) \in \\ [Pa_M(Y)], \\ pa_{O(M)} \subset pa_M}} |p_{O(M)}(y_i|pa_{O(M)}(Y)) - p_M(y_i|pa_M(Y))|$$

Given the $Y$ corresponding to the equation above, let $Pa_M(Y) = \{A_1...A_g..A_{g+h}\}$ and $Pa_{O(M)}(Y) = \{A_1...A_g\}$. Since $P_{O(M)}(y_i|pa_{O(M)}(Y))$ is the marginalization of all $P_M(y_i|pa_M(Y))$, we have:

$$|p_{O(M)}(y_i|pa_{O(M)}(Y)) - p_M(y_i|pa_M(Y))| \leq \max_{\substack{pa_M(Y)_1, \\ pa_M(Y)_2 \\ \in [Pa_M(Y)], \\ s.t. pa_{O(M)} \subset \\ pa_M(Y)_1 \cap pa_M(Y)_2}} |p_M(y_i|pa_M(Y)_1) - p_M(y_i|pa_M(Y)_2)|$$

$$< \sum_{j=1}^{j=h} e_M(A_{g+j} \to Y)$$

If $\sum_{j=1}^{j=h} e_M(A_{g+j} \to Y) < \delta/2$, then we have:

$$d[O(M), M] < \delta/2$$

For all such $M$, there is a $N_1$, such that for any $n > N_1$:

$$P_M^n(d[\hat{O}(M), M] > \delta | C(n, M) = O)$$
$$\leq P_M^n(d[\hat{O}(M), O(M)] + d[O(M), M] > \delta | C(n, M) = O)$$
$$\leq P_M^n(d[\hat{O}(M), O(M)] > \delta/2 | C(n, M) = O) < \epsilon$$

If $\sum_{j=1}^{j=h} e_M(A_{g+j} \to Y) \geq \delta/2$, then there is at least an $w \in \{1, 2...h\}$, $s.t.$ $e_M(A_{g+w} \to Y) > \frac{\delta}{2h}$. By Lemma 2:

$$\epsilon_{A_{g+w},Y|\mathbf{U}} \geq T^{|\mathbf{U}|+1} e_M(A_{g+w} \to Y) > T^{|\mathbf{U}|+1} \frac{\delta}{2h}.$$

where the $\mathbf{U} \cup \{A_{g+w}, Y\}$ is some ancestral set not containing any descendant of $Y$.

Since the density estimation does not turn "unknown", we know that in step 5 of $VCSGS$ the test of $\epsilon_{A_{g+w},Y|U} = 0$ returns 0 while $\epsilon_{A_{g+w},Y|U} \geq T^{|\mathbf{U}|+1}\frac{\delta}{2h}$. Since the test is uniformly consistent, it follows that there is a sample size $N_2$ such that

$$P_M^{N_2}(\epsilon_{A_{g+w},Y|U} = 0) < \lambda$$

for any $n > N_2$ and therefore for all such $\mathbf{M}$,

$$P_M^n(d[\hat{O}(M), M] > \delta | C(n, M) = O) < \lambda$$

Let $N = max(N_1, N_2)$, for $n > N$,

$$\lim_{n \to \infty} \sup_{M \in \Psi_3} P_M^n(d[\hat{O}(M), M] > \delta | C(n, M) = O) P_M^n(C(n, M) = O)$$
$$\leq \lim_{n \to \infty} \sup_{M \in \Psi_3} P_M^n(d[\hat{O}(M), M] > \delta | C(n, M) = O) = 0$$

∎

## 4. Discussion

We have shown that there is a uniformly consistent estimator of causal structure and some causal effects for nonparametric distributions under the **$k$-Triangle-Faithfulness Assumption**, which is sometimes stronger than the Faithfulness Assumption and weaker than the Strong Faithfulness Assumption. There are a number of open questions, such as whether the causal sufficiency assumption can be relaxed, so we allow the existence of latent variables and whether there are similar results under assumptions weaker than the Causal Faithfulness Assumption, such as the Sparsest Markov Representation Assumption (Solus et al.,2016).

## Acknowledgments

We thank a bunch of people and funding agency.

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

## Appendix
The appendix contains

- Proof of Lemma 2;

- Proof of Lemma 6;

- Proof of making error in *kind I* uniformly converges to zero;

- Proof of making error in *kind III* uniformly converges to zero;

## Lemma 2
*Given an ancestral set $\mathbf{A} \subset \mathbf{V}$ that contains the parents of $Y$ but not $Y$:*
*If $X$ is a parent of $Y$:*

$$T^{|\mathbf{A}|} e(X \to Y) \leq \epsilon_{X,Y|\mathbf{A}\backslash\{X\}} \leq e(X \to Y)$$

**Proof** Upper bound:

$$\epsilon_{X,Y|\mathbf{A}\backslash\{X\}} = \int_{\mathbf{A}\backslash\{X\}} \int_X \int_Y |p_{x,y,\mathbf{a}\backslash\{x\}} - p_{y|\mathbf{a}\backslash\{x\}} p_{x|\mathbf{a}\backslash\{x\}} p_{\mathbf{a}\backslash\{x\}}| dy dx d\mathbf{a} \backslash \{x\} \quad (5)$$

$$= \int_{\mathbf{A}} p_{\mathbf{a}} ||p_{y|x,\mathbf{A}\backslash\{X\}} - p_{y|\mathbf{A}\backslash\{X\}}||_1 d\mathbf{a} \quad (6)$$

$$\leq \mathbb{E}_{\mathbf{A}\sim p_{\mathbf{A}}}[e(X \to Y)] \quad (7)$$

$$= e(X \to Y) \quad (8)$$

Lower bound:

$$\epsilon_{X,Y|\mathbf{A}\backslash\{X\}} = \int_{\mathbf{A}\backslash\{X\}} \int_X \int_Y |p_{x,y|\mathbf{a}\backslash\{x\}} - p_{y|\mathbf{a}\backslash\{x\}} p_{x|\mathbf{a}\backslash\{x\}}| dy dx d\mathbf{a} \backslash \{x\} \quad (9)$$

$$= \int_{\mathbf{A}\backslash\{X\}} p_{\mathbf{a}\backslash\{x\}} \int_x p_{x|\mathbf{a}\backslash\{x\}} ||p_{Y|x,\mathbf{a}\backslash\{x\}} - p_{Y|\mathbf{a}\backslash\{x\}}||_1 dx d\mathbf{a} \backslash \{x\} \quad (10)$$

$$\geq T^{|\mathbf{A}|} e(X \to Y) \quad (11)$$

The step (11) is derived using a direct conclusion from NZ(T):
for any $\mathbf{V} \supset \mathbf{W} = \{W_1, W_2...W_n\}$, by the Chain Rule: $p_{\mathbf{W}} = \prod_{i=1}^n p_{W_i|W_{i+1}...W_n} \geq T^n$ ∎

**lemma 6** *If $\hat{p}(Y = y, Pa(Y) = pa(Y))$ and $\hat{p}(Pa(Y) = pa(Y))$ are uniformly consistent estimators of $p(Y = y, Pa(Y) = pa(Y))$ and $p(Pa(Y) = pa(Y))$, then*

$$\hat{p}(y|Pa(Y) = pa(Y)) = \frac{\hat{p}(Y = y, Pa(Y) = pa(Y))}{\hat{p}(Pa(Y) = pa(Y))}$$

*is a uniformly consistent estimator for $p(y|Pa(Y) = pa(Y))$*
**Proof** Recall that:
for any $\lambda < 1, sup_{P \in \mathcal{P}_{TV(L)}} P^n(||\hat{p}_h(x) - p(x)||_\infty \leq f(n,\lambda) = \mathcal{O}((\frac{\log n}{n})^{\frac{1}{2+d}})) \geq 1 - \lambda \ (*)$

where $f(n, \lambda)$ is continuous and monotonically decreasing wrt $n$ and $\lambda$ and $h \propto n^{2/(2+d)}$ (the number of bins) where $d$ is the dimentionality of the $x$. For instance, $d = |Pa(Y)| + 1$ when estimating $P(Y, Pa(Y))$.

For any $\delta > 0$, $f(n, \lambda) = \delta$ entails that $sup_{P \in \mathcal{P}_{[0,1],TV(L)}} P^n(||\hat{p}_h(x) - p(x)||_\infty > \delta) < \lambda$. By monotonicity of $f(n, \lambda)$, when $n > n_f$ s.t. $f(n_f, \lambda) \leq \delta$, $sup_{P \in \mathcal{P}_{[0,1],TV(L)}} P^n(||\hat{p}_h(x) - p(x)||_\infty > \delta) < \lambda$.

Let $d = |Pa(Y)|$, notice that for any $\lambda$, with probability at least $1 - \lambda$,[2][3]

$$|\frac{\hat{p}(Y = y, Pa(Y) = pa(Y))}{\hat{p}(Pa(Y) = pa(Y))} - p(y|Pa(Y) = pa(Y))| \tag{12}$$

$$= \frac{1}{\hat{p}(Pa(Y) = pa(Y))}|\hat{p}(Y = y, Pa(Y) = pa(Y)) - p(y|Pa(Y) = pa(Y))\hat{p}(Pa(Y) = pa(Y))| \tag{13}$$

$$\leq \frac{|p(Y = y, Pa(Y) = pa(Y)) + \mathcal{O}((\frac{\log n}{n})^{\frac{1}{3+d}}) - p(y|Pa(Y) = pa(Y))(p(Pa(Y) = pa(Y)) - \mathcal{O}((\frac{\log n}{n})^{\frac{1}{2+d}}))|}{T^d} \tag{14}$$

$$\leq \frac{1}{T^d}|\mathcal{O}\left((\frac{\log n}{n})^{\frac{1}{3+d}}\right) + \mathcal{O}\left((\frac{\log n}{n})^{\frac{1}{2+d}}\right)| \tag{15}$$

$$= \mathcal{O}\left((\frac{\log n}{n})^{\frac{1}{3+d}}\right) \tag{16}$$

Step (14) is derived by $(*)$ and the fact that the estimation result can only be valid if it satisfies TV smoothness(L)); step (15) is derived because $p(y|Pa(Y) = pa(Y))$ is upper bounded by $(1 + L|Pa(Y)|)$ by TV smoothness.

We have:

$$sup_{P \in \mathcal{P}_{TV(L)}} P^n \left[ |\frac{\hat{p}(Y = y, Pa(Y) = pa(Y))}{\hat{p}(Pa(Y) = pa(Y))} - p(y|Pa(Y) = pa(Y))| \leq O\left((\frac{\log n}{n})^{\frac{1}{3+d}}\right) \right]$$

$$\geq 1 - \lambda$$

So $\hat{p}(y|Pa(Y) = pa(Y)) = \frac{\hat{p}(Y = y, Pa(Y) = pa(Y))}{\hat{p}(Pa(Y) = pa(Y))}$ is a uniform consistent estimator for $p(y|Pa(Y) = pa(Y))$ ∎

**Lemma 7** *(Spirtes and Zhang, 2014) Given causal sufficiency of the measured variables* **V***, the Causal Markov, $k$-Triangle-Faithfulness, TV smoothness(L) assumption and NZ(T) assumption:*[4]

$$\lim_{n \to \infty} \sup_{M \in \phi^{k,L,C}} P^n_M(C(n, M) \text{ errs in kind I}) = 0$$

**Proof** $C(n, M)$ has an adjacency not in $G_M$ only if some test of zero conditional dependence rejects its null hypothesis. Since uniformly consistent tests are used in VCSGS, for every $\lambda > 0$,

---

2. here we use $\hat{p}(x)$ instead of $\hat{p}_h(x)$ because $h$ is dependent on the dimension of $x$

3. Recall that **V** denotes the set of variables in the true graph

4. We've changed the notations of the original proof to match with the notations used in this paper.

for every test of zero conditional dependence $t_i$, there is a sample size $N_i$ such that for all $n > N_i$ the supremum (over $\phi^{k,L,C}$) of the probability of the test falsely rejecting its null hypothesis is less than $\lambda$. Given $\mathbf{V}$, there are only finitely many possible tests of zero conditional dependences. Thus, for every $\lambda > 0$, there is a sample size $N$ such that for all $n > N$, the supremum (over $\phi^{k,L,C}$) of the probability of any of the tests falsely rejecting its null hypothesis is less than $\lambda$. The lemma then follows.

∎

**Lemma 8** *(Spirtes and Zhang, 2014) Given causal sufficiency of the measured variables* $\mathbf{V}$, *the Causal Markov,* $k$-*Triangle-Faithfulness, TV smoothness(L) assumption and NZ(T) assumption:*

$$\lim_{n \to \infty} \sup_{M \in \phi^{k,L,C}} P_M^n(C(n,M) \text{ errs in kind III}) = 0$$

**Proof**  Given that all the adjacencies and marked noncolliders in $C(n,M)$ are correct, there is a mistaken orientation if and only if there is an unshielded collider in $C(n,M)$ which is not a collider in $G_M$, for the other orientation rules in step 4 in $VCSGS$ would not lead to any mistaken orientation if all the unshielded colliders were correct. Thus, $C(n,M)$ errs in *kind III* only if there is a noncollider $X, Y, Z$ in $G_M$ that is marked as an unshielded collider in $C(n,M)$.

There are then two cases to consider:

$(III.1)$ $C(n,M)$ contains an unshielded collider that is an unshielded noncollider in $G_M$, and $(III.2)$ $C(n,M)$ contains an unshielded collider that is a shielded noncollider in $G_M$. The argument for case $(III.1)$ is extremely similar to that for $(I)$ in the proof of *kind II* (Lemma 3), and the argument for case $(III.2)$ is extremely similar to that for $(II)$ in the proof of *kind II*.  ∎

