# OpenReview forum: "A Uniformly Consistent Estimator of non-Gaussian Causal Effects Under the $k$-Triangle-Faithfulness Assumption"
_cclear.cc/CLeaR/2022/Conference — CLeaR 2022 Poster_

### Official Review · Reviewer_AaJn · 2021-11-08

**Confidence:** 3
**Overall Score:** 5

**Main Review:**

Main comments
-	My main concern with this paper is readability: I found it very hard to extract the information. After the acceptable introduction, the paper dives into a series of lemmas and proofs, where it is easy to lose the bigger picture. Moreover, the notation is not always consistent and the display of the main equation arrays is too much croweded to the left, so hard to read. I will give several examples below. Because of this, I think the current form of the paper is not acceptable for publication.
-	There are several gaps in the proofs (claiming that this step is easy or similar to some other proof), where it is hard to assess the correctness of the proof.
-	I don’t think that the results presented will have a great impact on the field. They more seem like variations of existing approaches.
-	Could the authors mention on the practical applicability of the algorithms (e.g. computational complexity)?

Detailed Comments:
-	The claims and outlooks in the introduction are not well matched in the rest of the paper. E.g. the term “Generalized k-Triangle Faithfulness” is not mentioned once again after the introduction. I suggest a much more rigorous format with careful use of formal Definitions for new assumptions and an algorithmic pseudo-code for algorithms. Moreover, the main findings should be clearly marked with Theorems and the very technical lemmas (e.g. Lemma 4, which is presented within a proof) should be moved to the appendix or supplementary material.
-	I is not clear if the two algorithms proposed are new in themselves or if the consistency result is new.
-	Delete sentence at the end of Introduction: “This is where the content of your paper goes”
-	Ch 2.1., end of first paragraph: “and Y is a descendant of X” (X and Y are in the wrong order in the current text)
-	Ch 2.2: “its its”
-	Ch 2.2, end of page 2: “the set of X variables is jointly standard Gaussian”; I am not sure if “standard” applies here, it should probably be moved to the error term?
-	Ch 2.2: spelling in “week anyywhere”
-	Ch 2.2: Does the “K” in the inequalities and the “k” in the last sentence of ch 2.2 mean the same value? If yes, please use same symbol, if no, please explain how the “k-triangle-Faithfulness” relates to the two versions of “k”.
-	Ch 2.3, step 3: The term “unshielded” was not defined (although it is pretty clear what you mean)
-	Ch 2.3, step 4a: I am wondering if you should add “unshielded” to “the triple (X,Y,Z)” ?
-	Ch 2.3, step 5: “apparently non-adjacent” was not defined and does not occur in the algorithm before
-	Ch 3, second bullet point: In the subscript you use both “Y” and “Y=y”. Before you also just used the subscript “y”. Please stay consistent to improve readability.
-	Ch 3: What does the “TV” in the “TV Smoothness” mean?
-	Ch 3: What norm are you referring to in “\leq (1 + L |Pa(Y)|)” ? Should that really be a capital “P” in “Pa” or rather “pa”?
-	Ch 3.1: Can you give the rationale behind using the proposed definition of epsilon? Since it characterizes conditional dependence given A, I would have thought of: p_{X,Y|A} vs. p_{X|A}*p_{Y|A}. Why are you using the joint and not the conditional probability?
-	Ch 3.1, p.6: Typo in math expression in 3rd paragraph: “lrangle”
-	Ch 3.1, p.6: “size size”
-	Ch 3.1, list item 2: “G_M” instead of “G_m”
-	Ch 3.1: “no undirected edge connecting Y AND X” (I think “and X” is missing)
-	Ch 3.1: “intervening ON X” (“on” is missing)
-	Ch 3.1: “Since the proofs of kind I and kind III are almost the same … we are only going to prove kind II here” – this is not obvious to me.
-	The proof of Lemma 2 is hard to check: Please add some details throughout the proof on the main strategy of the proof. E.g., you prove by contradiction, but in the end there is no clear mentioning of the contradiction you achieved. Moreover, the reference “Lemma 3.1” should be “Lemma 1”. Moreover, the notation of “K” in the denominator is not consistent.
-	Edge Estimation Algorithm: Please set this apart clearly from the main text. I suggest pseudo-code and an algorithm display.
-	The definition of “conditional probability distance” is hard to read. Is there a way to simplify the notation?
-	P.9: The proof starts without an easy to see claim. Since this seems to be one of the main results, you could perhaps reformulate this as a Theorem (ideally you name all your main theorems) to help the reader to filter out the main ideas from the technical details.
-	The references are not complete. At least “Kalisch and Bühlmann (2007)” appears in the main text but not in the reference section.


**Summary:**

The authors present a result on uniform consistency for estimating causal structures in the case of of smooth distributions. By this, the authors extend existing results mainly restricted to linear Gaussian models. The way the achieve this is by replacing assumptions. Moreover, they present an algorithm for causal structure learning (VCSGS) and an algorithm for estimating causal effect size (EEA) and prove uniform consistency in the inference of the causal structure.

---

> ### Author Response · Authors · 2021-12-03
> **Response to comments and suggestions**
>
> Dear Reviewers:
>
> We thank the reviewers for their comments and suggestions. We will fix the typos and change the formatting of the proofs to make them more readable. We will also correct mistake in the algorithm, and move the proofs of the lemmas to the appendix, to open room for more discussion of the overall strategy, and how the result relates to previous research.  For your questions about this paper, please kindly find my response below:
>
>
> * I don’t think that the results presented will have a great impact on the field. They more seem like variations of existing approaches.
>
>
> **Respond**: The result of our work shows that given uniform consistency of nonparametric conditional independence tests (which itself requires a smoothness assumption), we can reach uniform consistency of estimating causal structures and effects without making parametric assumptions. The only uniform consistency results for causal search and estimation algorithms that we know of (Spirtes and Zhang, Kalisch and Buhlmann) assume linear Gaussian models. Since uniform consistency allows one to give probabilistic bounds on the size of errors given a sample size as a function of k and the smoothness parameter, this gives some small sample results for algorithms that only currently have large sample guarantees.
>
>
> * Could the authors mention on the practical applicability of the algorithms (e.g. computational complexity)?
>
>
> **Respond**:  We will provide the computational complexity of the algorithms and the convergence rate of estimation.
>
>
> * not clear if the two algorithms proposed are new in themselves or if the consistency result is new.
>
>
> **Respond**: The consistency result is new.
>
>
> * Ch 3.1: Can you give the rationale behind using the proposed definition of epsilon? Since it characterizes conditional dependence given A, I would have thought of: p_{X,Y|A} vs. p_{X|A}*p_{Y|A}. Why are you using the joint and not the conditional probability?
>
>
> **Respond**: We want to characterize the $p_\{X,Y|A\}$ vs. $p_\{X|A\}$*$p_\{Y|A\}$ over all values of X, Y and A.  You can think about our definition of epsilon as a general characterization of all $|p_\{X,Y|A\} -p_\{X|A\}*p_\{Y|A\}|$ over all values of X, Y and A
>
>
> * Ch 3.1: “Since the proofs of kind I and kind III are almost the same … we are only going to prove kind II here” – this is not obvious to me.
>
>
> **Respond**:  We will move the proofs of the lemmas into the appendix and provide more details. The only additions that we need to make to the proofs of kind I and III are minor modifications of the corresponding proofs in Spirtes and Zhang.

---

> > ### Comment · Reviewer_AaJn · 2021-12-07
> > **Update**
> >
> > I would like to thank the authors for their response. After digesting their answers, I see the situation in the following way:
> > - The result in itself seems to be indeed potentially interesting to the community; therefore, I am willing to increase my score
> > - I trust that it is possible to improve the readability of the paper as the authors indicate. However, I think this will be quite challenging and therefore my final score is still marginal.

---

### Official Review · Reviewer_25b4 · 2021-11-23

**Confidence:** 3
**Overall Score:** 5

**Main Review:**

The authors consider the existing VCSGS algorithm and show that under suitable assumptions, it leads to uniformly-consistent estimation of causal strengths.

Although the technical content and contribution seem interesting, the paper needs to undergo a significant revision for the reader to be able to appreciate the material. I also have some comments about the content below and would be happy to hear from the authors.

"k-Triangle-Faithfulness Assumption is a strictly weaker assumption that avoids some implausible implications of the Strong Causal Faithfulness Assumption and also allows for uniformly consistent estimates of Markov Equivalence Classes (in a weakened sense), and of identifiable causal effects."
Please add a citation for the k-Triangle Faithfulness assumption.

Please delete the following at the end of page 1:
"This is where the content of your paper goes."

"Kalisch and Bu ̈hlmann (2007) showed that such uniform consistency is achieved by the PC algorithm if the underlying DAG is sparse relative to the sample size"
Could you explain what this means? Sparsity is typically relative to number of nodes. Is this result independent from the number of nodes?

"It has the implausible consequence that it puts a lower bound on the strength of edges, since a very weak edge entails a very weak partial correlation."
I don't understand this statement. It's an assumption and there are many models where such an assumption would be true. Why is this called implausible?

Could you define "k-Faithfulness Assumption"?

Is there a distinction between lowercase k and capital K?

"edges to be week anyywhere in a graph"
two typos

Rule 4(c) seems wrong. Checking the original version from https://arxiv.org/pdf/1502.00829.pdf, in S4.iii, we see that X->Y<-Z is the condition, not X->Y->Z.

Please use \lim and \sup for better visibility of formulas.

There are also typos in latex, such as "lrangle".

In terms of organization, I would add proof sketches to the main paper and only keep those in the main text. Moving the proofs will lead to better readability even if the main paper remains much shorter than 12 pages. That is not a problem.

"while holding everything else constant"
Please specify that only parents of Y are conditioned on. Not descendants.

max->\max

Also I suggest using \limits to move limits to right under the operator for better visibility.

Could you comment on why max is chosen for defining the edge strength rather than, for example, average? How will the rest of the approach be affected by such changes to the definition? Is the approach robust to this?

"Given the TV smoothness(L), p is continuous"
I am not sure if the terminology is accurate here. A discrete p will never be continuous.

I am not sure if I follow the reasoning behind the derivation before Section 3.1 for p_y|Pa(Y)... This looks like a very loose bound and the connection to the narrative (why do we need this derivation?) is not clear to me.

"Then the test is uniformly consistent w.r.t. a set of distributions Ω for"
"for" should be replaced with "if" I believe.

ani.i.d. sample of size size n->an i.i.d. sample of size n

The equations at the end of page 7 are really hard to parse due to abrupt and unaligned line breaks. Please consider using a controlled align environment.

Proof of Theorem 3:
"Since we have proved that the probability for C(n,M) to make any of the three kinds of mistakes uniformly converges to 0, the theorem directly follows."
I would appreciate seeing more details on this step. I believe the reasoning is that any error is either type I OR type II OR type III, correct? But this needs to be considered in the lim sup and at least a short statement detailing this would be nice to have.

"are true inM"->"are true in M"

Please write down the theorem for which the proof is given in page 9 explicitly.

"(the number of bins)"
What is a bin? This is not explained as far as I can see.

Again, it is very very hard to parse the equations/proofs in page 10 due to the way equations are presented. Please consider using align and breaking into newlines for better readability. Same for page 11.

Please use \mathcal{O} instead of O for big-oh notation and use \left(\right) instead of () for large parantheses.

I believe I missed the part where the following statement became relevant in the later section:
"Notice that missing an edge is not listed as an mistake here, and we are going to prove later that the estimation of causal influence can still be used to correctly predict the effect of intervention even if the algorithm misses edges."

I believe the analysis looks mostly fine but the paper suffers a lot from the clarity of presentation. As recommended above, moving proofs to the appendix and using main paper to convey intuitions behind the theorems and proofs would help greatly and make room for more discussion about connections to previous work.

**Summary:**

Review report

---

> ### Author Response · Authors · 2021-12-03
> **Response to comments and suggestions**
>
> Dear Reviewers:
>
> Thank you for your comments and suggestions. We will fix the typos and change the formatting of the proofs to make them more readable.  For your questions about this paper, please kindly find my response below:
> * "Kalisch and Bu ̈hlmann (2007) showed that such uniform consistency is achieved by the PC algorithm if the underlying DAG is sparse relative to the sample size"
> Could you explain what this means? Sparsity is typically relative to number of nodes. Is this result independent from the number of nodes?
>
> **Respond**:  This result is not independent from the number of nodes. Kalisch and Bu ̈hlmann assumped that the maximum number of neighbors, q, in the DAG, grows as q = O(n^{1-b}), with n being the number of nodes and 0<b<1.
>
> * "It has the implausible consequence that it puts a lower bound on the strength of edges, since a very weak edge entails a very weak partial correlation."
> I don't understand this statement. It's an assumption and there are many models where such an assumption would be true. Why is this called implausible?
>
> **Respond**: A better way for us to phrase this is that it is often implicitly assumed that there may be many weak causes of a variable, but even if they exist, that they can be safely ignored in estimating causal effects. The k-triangle faithfulness assumption basically formalizes that assumption, whereas the Kalisch-Buhlmann results require the stronger assumption that there are no such weak causes. That assumption is not necessary to obtain uniform consistency, and limits the applicability of their results.
>
> * Could you comment on why max is chosen for defining the edge strength rather than, for example, average? How will the rest of the approach be affected by such changes to the definition? Is the approach robust to this?
>
> **Respond**: Yes, the average can also be chosen to define the edge strength.  The rest of the approach will be changed slightly, and the approach is robust to this. We choose the max because we are particularly interested in weak edges, and if an edge is weak, the maximum of change on effect when changing the state of cause is small.
>
> * "Given the TV smoothness(L), p is continuous"
> I am not sure if the terminology is accurate here. A discrete p will never be continuous.
>
> **Respond**: Here by continuous I mean that if a and a’ are arbitrarily close, p_y conditioning on A =a and A=a’ are arbitrarily close.
>
> * I am not sure if I follow the reasoning behind the derivation before Section 3.1 for p_y|Pa(Y)... This looks like a very loose bound and the connection to the narrative (why do we need this derivation?) is not clear to me.
>
> **Respond**: We just want to show that p_y|Pa(Y) cannot be infinitely large; it is used later in the proof of lemma 4.
>
> * Proof of Theorem 3: "Since we have proved that the probability for C(n,M) to make any of the three kinds of mistakes uniformly converges to 0, the theorem directly follows." I would appreciate seeing more details on this step. I believe the reasoning is that any error is either type I OR type II OR type III, correct? But this needs to be considered in the lim sup and at least a short statement detailing this would be nice to have.
>
> **Respond**: The reasoning is correct.  We will provide more details.
>
> * "(the number of bins)" What is a bin? This is not explained as far as I can see.
>
>
> **Respond**:  To test whether x, y are independent conditioning on z, the conditional independence test we used in this work separates the z into bins.  We will explain it in the paper.
>
> * I believe I missed the part where the following statement became relevant in the later section:
> "Notice that missing an edge is not listed as a mistake here, and we are going to prove later that the estimation of causal influence can still be used to correctly predict the effect of intervention even if the algorithm misses edges."
>
> **Respond**:  Here we want to say that, an edge is only going to be missed if it is a weak edge; and if such an edge is missed, we still can achieve uniform consistency when estimating p_y|Pa’(Y) where here Pa’(Y) is the parent of Y except those connected to Y through the missed, weak edges.

---

### Official Review · Reviewer_RX9d · 2021-11-26

**Confidence:** 3
**Overall Score:** 8

**Main Review:**

The paper provides a non-parametric version of the k-Triangle-faithfulness assumptions and shows that, under that assumption, there is a uniformly consistent procedure for identifying the Markov equivalence class of a causal model.   Extending the k-faithfullness assumption to non-linear models is a substantial step, I think, and is worth of publication.  I was unable to check carefully all of the arguments in the reviewing time given, but I believe general line of argument is correct.  However, the paper is filled with typographical errors and with poor typesetting.  I don't know how to evaluate the importance of those errors as a reviewer.

**Summary:**

The paper provides a non-parametric version of the k-Triangle-faithfulness assumptions and shows that, under that assumption, there is a uniformly consistent procedure for identifying the Markov equivalence class of a causal model.

---

### Decision · Program_Chairs · 2022-01-12

**Decision:**

Accept (Poster)

**Comment:**

This is a theory paper that tries to generalize the k-Triangle-Faithfulness Assumption to non-Gaussian non-parametric cases.  Though the reviewers found the result having interesting things, the readability and clarity are not sufficiently good. I want the authors to reflect all the points raised by the reviewers.